# What Do You Need to Know? A Systematic Review and Research Agenda on Neuromarketing Discipline

Prakash Singh , Ibrahim Alhassan and Lama Khoshaim *

E-Commerce Department, College of Administrative and Financial Sciences, Saudi Electronic University, Riyadh 11673, Saudi Arabia; p.kishore@seu.edu.sa (P.S.); ialhassan@seu.edu.sa (I.A.)
* Correspondence: l.khoshaim@seu.edu.sa

**Abstract:** The neuromarketing phenomenon has led to a makeover in the marketing area, and its application in the business world has generated a better insight into understanding diverse consumer behavior. This comprehensive study delves into the multifaceted world of neuromarketing. Employing a systematic literature review approach and reviewing 51 articles from the Web of Science database, the study aims to provide a holistic view of the neuromarketing field, offering valuable insights and directions for future research and practical application in the business and academic world. The key results of this study are classified into six distinct research themes: 1. Evolution of Neuromarketing, 2. Neuromarketing Definitions and Neuromarketing Tools, 3. Neuromarketing in Practice, 4. Social Value of Neuromarketing, 5. Neuromarketing and Consumer Behavior, and 6. Neuromarketing for Sustainable Business Development. The results of the study are based on the 4W framework (What, Where, Why, and How) and the TCCM framework (Themes, Contexts, Characteristics, and Methodology). The study underscores the significance of neuromarketing to businesses and serves as a provocative call to action for businesses to study its potential which traditional marketing techniques may overlook. This study is notable for its investigation of theoretical evolution, definitions, tools, practices, social value, and influence on consumer behavior within the discipline. Its contribution unfolds to sustainable business development, where neuromarketing can facilitate sustainable products and practices by understanding consumer behavior. This study presents valuable insights and sets the stage for future research through theoretical advancements in neuromarketing. It further acts as a foundational resource for marketers and researchers in augmenting their theoretical and practical acumen.

**Keywords:** Businesses; Consumer behavior; Neuromarketing; Systematic Literature review; TCCM framework; Web of Science; 4W framework

## 1. Introduction

Consumers play a significant role in the conception of sustainable business development. The preferences made by consumers while procuring products impact businesses in deciding which products to produce and how they will be delivered [1]. One of the leading concerns in today's business market is what leads consumers to decide on one product for consumption instead of another or why consumers relate to a specific product. Subsequently, businesses are curious about how the brain responds and signals to consumers' buying decision processes [2,3]. To understand the progressively multi-faceted consumer buying decision process and consumption environment, contemporary marketing researchers, scholars, and academicians have started to learn the leading drivers of consumers' buying decision process from a multi-disciplinary perspective. Furthermore, the marketing domain has transformed substantially, accustoming to the multi-dimensional assessment of consumers' preferences by encompassing and promising ideas, concepts, and practices resulting from diverse disciplines, namely, sociology, psychology, anthropology, and a more newly evolved marketing domain, i.e., neuromarketing. At Harvard University,

"Neuromarketing" as an idea was created by therapists in 1990 [4,5]. However, in 2002, Ale Smidts, a Dutch organizational theorist who was a marketing professor, presented the term "Neuromarketing" [6]. Neuromarketing as a research domain is rapidly budding and is assisting businesses to create more incredible acumen in consumer behavior, particularly their physiological actions. This further contributes deliberately to companies' sustainability [5–8]. Physiological actions can enlighten consumers' responses and aid contemporary businesses in diagnosing the buying decision process of consumers by adopting neuromarketing practices [1]. The endeavor to forecast consumer behavior from physiological data can be traced back to prior studies [9–11]. Thus, marketing has grown from learning consumers' diverse behaviors to delving into consumers' brain retorts about marketing stimuli, i.e., recognized as the phenomenon of "Neuromarketing" [2,3].

"Neuromarketing" is part of the "Neuroscience" research domain that aims to anticipate the consumer's behavior through the cerebrum's basic actions and responses. It is demarcated as the field of study of the cerebral apparatus to comprehend consumers' behavior to advance a business's marketing strategies [6]. Moreover, neuromarketing is a newly evolving field in the marketing domain that judges consumers' brain responses using marketing actions over diverse medical instruments. It ensures an effective marketing process for sustainable development [12]. Even though neuromarketing can potentially deliver value to the contemporary consumer research domain in several ways, marketers and businesses are still not employing the concept of neuroscience in marketing to accomplish sustainable development through the profusely captivating benefit of neuromarketing in practice [8,13]. Neuromarketing offers innovative methods that, if incorporated into other business practices, may stimulate and broaden marketing strategies' outcomes, resulting in sustainable development [8,13,14]. The expansion of electronic commerce (E-commerce) has given businesses a significant prospect to market their products and services over the Internet. In this endeavor, neuromarketing plays a crucial part in understanding how customers behave when they shop online [15]. However, E-commerce platforms have embraced a neuromarketing framework to predict the future preferences of consumers during the buying process [16]. Contemporary businesses can be stimulated to advance market research by employing neuromarketing tools in their practices to understand diverse consumer responses better [14,17]. Contemplating the various potentials of neuromarketing, several extant works of literature have recorded precise production in this area of research. Amongst these, one study used a bibliometric approach to detect neuromarketing tools [18]. There has been extensive research on the scope and effectiveness of neuromarketing beyond its commercial applications through systematic literature reviews and empirical methods. For instance, de Oliveira et al. [19] studied the scope of neuromarketing, while Fugate [20] delved into its research practices. Fortunato et al. [21] explored the techniques and applications of neuromarketing in businesses, whereas Bakardjieva and Kimmel [22] investigated the effectiveness of marketing-based services using neuromarketing research. Meanwhile, Pop and Lorga [23] illustrated the basic process of neuromarketing, and Alsharif et al. [24] analyzed the tools employed in this field. Moreover, Alsharif et al. [25] explored consumers' subconscious behavior in biomedical technology while delving into how advertising works in human minds was studied by [26]. These stated studies employed a systematic literature review approach. However, Fayaz and Nawaz [27] used a bibliometric analysis approach to study neuromarketing, whereas Kotler et al. [28] used a conceptual approach to study the neurodynamic state of the human brain, and conscious minds were analyzed using a conceptual approach [29]. Furthermore, Alsharif et al. [30] reviewed the application of neuromarketing tools in the marketing mix, whereas critical factors that impact neuromarketing implementation in practice were explored using a qualitative approach by [31]. However, a gap in the literature still exists as no study has employed the 4W framework and the TCCM framework using a systematic literature review approach, so this study aims to fill that gap so that the theoretical evolution of the neuromarketing phenomenon can be understood holistically. Hence, more systematic literature review studies must be carried out in contemporary periods.

Predominantly, a theme-based study from a theoretical perspective may offer acumens for future research directions. A rigorous SLR approach can augment theoretical investigations that support academics and practitioners in the long run [32]. The extant literature underlines the advancement of neuromarketing literature by exploring its current trends. It needs more intensity and consequently tends to deliver research gaps for further deliberation. However, studies on this research theme were steered initially in the early 1990s. Despite the numerous business gains of neuromarketing, the phenomenon is nascent, with little consideration from academics, scholars, and researchers [33]. Therefore, systematic investigation mapping would help acquire more insights into this. Accordingly, this study is intended to address the five research questions (**RQs**) as stated:

**RQ1:** What do we know about the neuromarketing phenomenon in an academic context?
**RQ2:** Where is the neuromarketing research happening in geographical and sector/industry contexts?
**RQ3:** Why should academicians, practitioners, and policymakers know more about neuromarketing?
**RQ4:** HoW was the neuromarketing research conducted in a geographical context?
**RQ5:** What future research directions and implications for academics and practitioners can be proposed?

Answering these stated research questions, this study will provide a thorough understanding of contemporary research studies on neuromarketing. Precisely, the research objectives (**ROs**) of the study are:

**RO1:** To investigate prevailing studies about neuromarketing across diverse disciplines.
**RO2:** To recognize what we know about the neuromarketing phenomenon and its development in the contemporary era.
**RO3:** To comprehend where these research works have been conducted and how they were conducted methodologically.
**RO4:** To propose the future research directions of the study and recommend academic and practical implications of the study.

The systematic literature review approach is the most suitable approach to align the diverse outlooks in the contemporary research domain. The present study is directed by robust approaches of conducting the SLR-based study employing the 4W (*What*, *Where*, *Why*, *HoW*) framework [34], the TCCM (*Theme* (*T*), *Contexts* (*C*), *Characteristics* (*C*), *Methodology* (*M*)) frameworks [35] along with the SLR approach [36] and the Preferred Reporting Items for Systematic Reviews and Meta-Analysis (PRISMA) protocol [37], concentrating on investigating key themes, key contexts, key characteristics, and critical methodologies employed in the preceding works of the literature. Hence, based on the above-stated critical points of differentiation, this study differs from existing studies. To the authors' knowledge, no prior work has mapped neuromarketing research production in the Web of Science database by extracting articles published up to 31 December 2022 as the publication period. Previous work has yet to employ the SLR approach, the PRISMA protocol, the 4W framework, and the TCCM framework in a single study. No prior work has selected the final sample size of articles by extracting articles listed under the SSCI category of the Web of Science index. In this line of the flow of research, this research article presents the methodology used in Section 2. Section 3 offers the review structure. Section 4 outlines results and discussion by employing the 4W and TCCM frameworks. Section 5 gives limitations, future research directions, and implications. Finally, the study's conclusion is summarized in Section 5.

## 2. Methodology Used

The present study employed the systematic literature review (SLR) approach widely applied in business and management research studies to have consent over distinct research [36]. The literature review approach is commonly undertaken to examine a multidisciplinary research study's contemporary implications and conceptual understanding.

Various literature review approaches are available to record and review the field of study. Scholars can be competent to attain their research objectives by tackling the research questions associated with their field of study. However, numerous literature review approaches, for example, meta-analyses [38,39], narrative reviews [40], integrative reviews [41], PRISMA [37], and systematic literature reviews [36] are available for conducting research studies. These literature review approaches might be employed to discover how to answer research questions and report the existing research gaps by thoroughly reviewing the key results. In addition to the SLR approach, this study used the PRISMA protocol to conduct a systematic review [37].

## 2.1. Justification of the Selected Research Approaches

This study selected the SLR approach over various available literature review approaches for diverse reasons. First, unlike the narrative review approach, the SLR approach is very scientific [36]. Then, the SLR approach facilitates the production of comprehensive inferences confining from a broad, apparent, and detailed approach that fosters replicability [42–45]. Lastly, the SLR approach promotes the attainment of the current research gaps and provides a clear understanding grounded on the prevailing review of existing studies by commissioning a quantitative system [46]. Furthermore, this approach thoroughly acknowledges a set of systematic processes and undoubtedly anticipates edging meticulous omission, primarily by endeavoring to detect, assess, and generate all pertinent works of literature [47]. Accordingly, it was found that the SLR approach is most suitable for undertaking the research objectives. The PRISMA protocol was also selected as it is grounded on well-defined systematic procedures. Researchers and scholars can ensure that their study is transparent and comprehensive and has coverage of systematic reviews by the full-text review of each selected article to sort the most eligible reports [38].

## 2.2. Research Protocol

Using the research protocol, we first reviewed the extant literature presented in Figure 1 to address the research questions. The research protocol employed is stated in the following sub-sections.

The Web of Science database was used to extract relevant research studies published up to 31 December 2022 as the publication period. The search string used to extract the most eligible articles on this research theme was "Neuromarketing", and 445 records were identified in the initial stage.

## 2.3. Inclusion and Exclusion Criteria

To ensure thoroughness and quality, "Articles" as a document type and "English" as a language were included to identify and select the most relevant articles. After this article screening, 230 articles were identified. Out of the 230 articles, based on open access restrictions, 106 articles were found eligible for the full-text review. Afterward, articles indexed in SCIE, A&HCI, and ESCI Web of Science indexing categories were excluded. In total, 59 articles indexed in the "SSCI" Web of Science category were selected for the review process.

## 2.4. Final Selection

A thorough full-text reading of all 59 selected articles was performed to identify the most eligible articles considering two parameters. First, the selected article is related to the research theme of neuromarketing. Secondly, the selected research article focuses on all the acknowledged research questions supporting the stated research objectives of this study. Finally, the 51 most eligible articles were selected as the final sample size. The PRISMA flow diagram stating the final sample selection protocol for the 51 most eligible articles is shown in Figure 1.

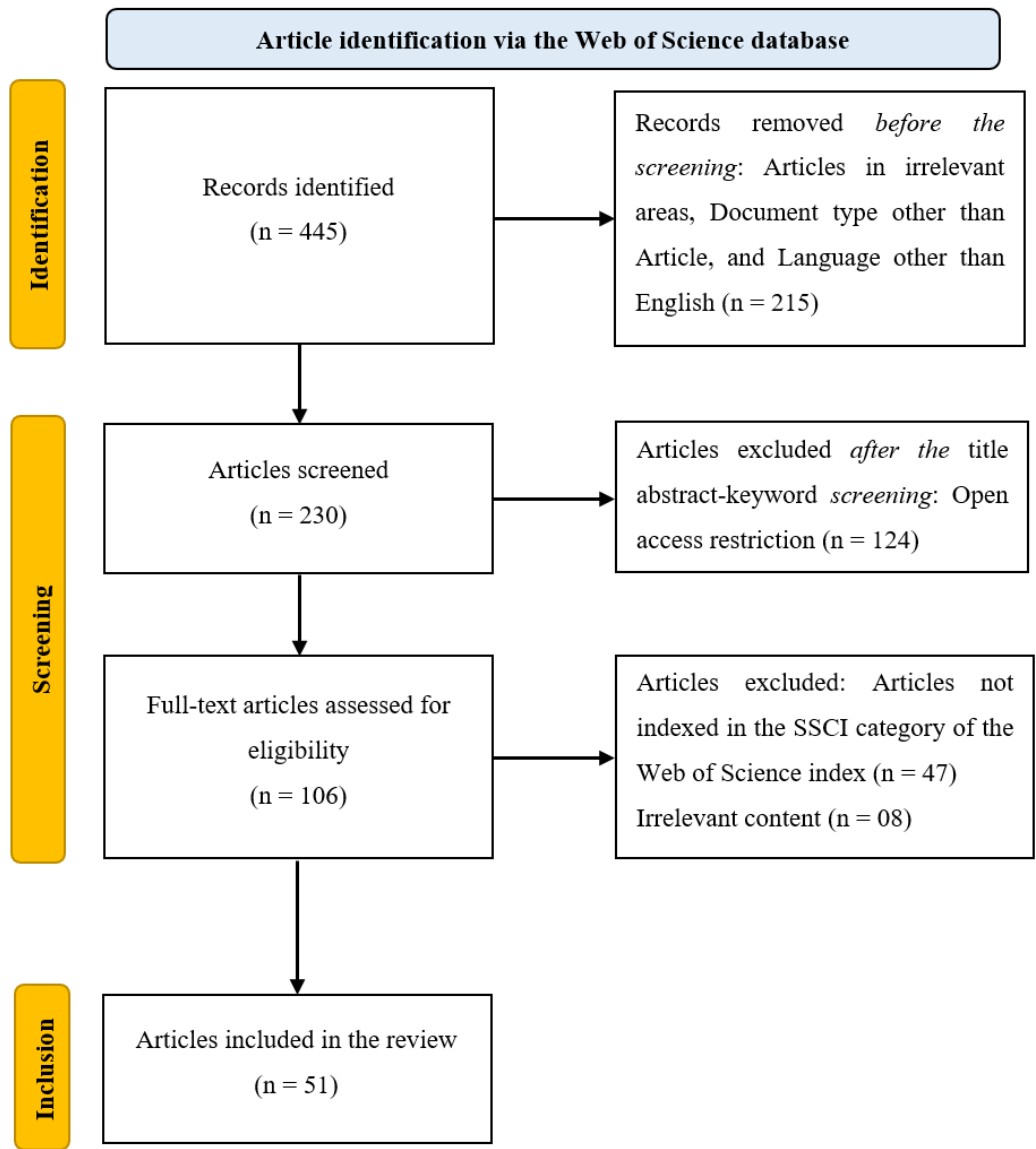

**Figure 1.** PRISMA flow diagram of the sample selection protocol.

*2.5. Data Extraction and Analysis*

The list of journals related to the 51 eligible articles was tabulated as shown in Table 1, which depicts the frequency of article occurrence in that journal indexed in the "SSCI" Web of Science index category. A comprehensive review of each article was conducted, and the next section of the study sums up the comprehensive review structure.

**Table 1.** List of Journals publishing research articles.

| S. No. | List of Journals | Frequency |
|:---:|:---|:---:|
| 1 | *Behavioral Sciences* | 03 |
| 2 | *Biosocieties* | 01 |
| 3 | *BMC Neurology* | 01 |
| 4 | *Brain Sciences* | 01 |
| 5 | *British Food Journal* | 01 |
| 6 | *Business History* | 01 |
| 7 | *Computational and Mathematical Methods in Medicine* | 01 |
| 8 | *Comunicar* | 01 |
| 9 | *Consumption Markets & Culture* | 01 |
| 10 | *Economic Computation and Economic Cybernetics Studies and Research* | 01 |

**Table 1.** *Cont.*

| S. No. | List of Journals | Frequency |
|:---:|:---|:---:|
| 11 | *Energies* | 01 |
| 12 | *European Journal of Marketing* | 01 |
| 13 | *European Research on Management and Business Economics* | 01 |
| 14 | *Frontiers in Human Neuroscience* | 01 |
| 15 | *Frontiers in Psychiatry* | 01 |
| 16 | *Frontiers in Psychology* | 14 |
| 17 | *Harvard Review of Psychiatry* | 01 |
| 18 | *Journal of Advertising Research* | 01 |
| 19 | *Journal of Brand Management* | 01 |
| 20 | *Journal of Marketing Management* | 01 |
| 21 | *Journal of Neuroscience Psychology and Economics* | 01 |
| 22 | *Marketing Intelligence & Planning* | 01 |
| 23 | *Mind, Brain, and Education* | 01 |
| 24 | *Neuroethics* | 01 |
| 25 | *Oeconomia Copernicana* | 01 |
| 26 | *Organizational Research Methods* | 01 |
| 27 | *Profesional De La Informacion* | 01 |
| 28 | *Science Technology & Human Values* | 01 |
| 29 | *Scientific Reports* | 01 |
| 30 | *Sensors and Materials* | 01 |
| 31 | *Society* | 01 |
| 32 | *Sustainability* | 04 |
| 33 | *Symmetry—Basel* Grand Total—51 | 01 |

## 3. Results and Discussion

Based on the literature review, we deduce that neuromarketing is not commonly well-defined in persistent statuses. The study's results revealed that the extant literature in neuromarketing had been fragmented into transversely diverse disciplines. To report these confines and to create theoretical lucidity, we start the discussion by addressing the aspect of 'What' in the 4W framework [34] to identify theoretical evidence that has occurred about the "neuromarketing" phenomenon. We categorized our findings into the following six themes, which link to the TCCM framework [35], to present the aspect of 'Theme (T)' development in the academic context.

### 3.1. Theme: What Do We Know about Neuromarketing in an Academic Context?

3.1.1. Evolution of Neuromarketing

In 1957, James Vicary, a marketing scholar, used "neuromarketing" in a movie theater in New Jersey to add up an automated shot to the single-frame projection, and this was acknowledged as the evolution of the term "Neuromarketing" [37]. Later, an Atlanta advertising company in the United States proposed this term in 2002, offering a substantial and appropriate birthdate [37,48]. However, the initial advent of the term "Neuromarketing" can be traced back to the middle of 2007 in the works [49,50]. Preceding this, pertinent marketing topics have been observed in a few works of literature in neuroeconomics [51–53]. Furthermore, in the neuroscientific commentaries about marketing practices, the term "neuromarketing" was also observed in the editorials [54]. Therefore, neuromarketing is an interdisciplinary research field [55,56].

3.1.2. Neuromarketing Definitions and Neuromarketing Tools

Neuromarketing is the study of the psychological assessment of consumers' sensory responses toward marketing communication. Neuromarketing aims to respond to marketing questions by exploring the human brain and its approach, engaging in better knowledge of how intuitive minds answer marketing activities. Likewise, neuromarketing delivers a place for investigation that cuts through various disciplines and sciences: neuroscience,

psychology, marketing, engineering, and economics [57–59]. In the neuromarketing domain, powerful tools are mainly built on electroencephalography (EEG), eye tracking, fMRI, facial expression assessment, etc. Furthermore, neuromarketing studies have demonstrated a gain in the neurosciences domain related to neuroimaging surrounding fMRI [60,61].

### 3.1.3. Neuromarketing in Practice

In the academic and commercial world, neuromarketing is a nascent study area. It engages the brain's action measurement equipment, brain imaging, and biometric techniques to grab consumers' brain responses to marketing inducements [62]. However, marketing study is at its crux, close to determining, learning, and forecasting behaviors of consumers in the marketplace. Including neuroscience practices in marketing studies can generate a sounder thought of the marketing practices' influence on consumers [7]. More precisely, neuromarketing has stimulated prodigious attention in the scientific study of the behaviors of consumers in the advertising industry. However, despite its expansion in the academic and commercial worlds, neuromarketing in practice is still very limited [63]. The practices of neuromarketing approaches may perhaps deliver marketers with novel information not reachable by traditional marketing study approaches [64]. The justification of neuromarketing practices lies in the hypothesis of understanding in what way the brains of consumers notice, process, and ponder [65]. In addition, neuromarketing tools can be employed in practices in diverse possible areas or sectors to understand the physiological actions of the consumer's brain and predict customer decision-making [66].

### 3.1.4. Social Value of Neuromarketing

Neuromarketing research can contribute to social value by understanding consumers' mental health and contributing to socially valuable results. However, when businesses, especially private companies, conduct neuromarketing research, research approaches and critical results are kept highly confidential, diminishing the social value of such research studies. Still, substandard neuromarketing research studies are frequently conducted, employing society for merely monetary gains, which can erode society and the community's confidence in such research studies [67]. As a term, neuromarketing is part of public debates about its negative social consequences [68]. In addition, the prime issue in front of neuromarketing applications is diverse social needs, predominantly when it originates in marketing communication, and it is complex to understand human brains [69]. However, to build healthy societies, several countries have been employing neuromarketing tools in practices to foster social welfare [70]. The stimulus of social information on consumer behavior offers the groundwork for research in the field of neuromarketing to uncover the secrets of consumer decision-making [71]. Despite the wide-ranging works of literature discussing verbal and non-verbal communication in politics, particularly during election campaigns, neuromarketing plays a key role. However, the study of how neuromarketing tools impact politics is still embryonic and requires detailed probing [72]. Hence, businesses must learn more about neuromarketing and consumer behavior for sustainable development.

### 3.1.5. Neuromarketing and Consumer Behavior

Predicting consumer buying decision behavior by employing a neuroscience approach to understanding the consumer mind is still a prominent issue encountered by global marketers due to technological disruptions and diverse needs and wants. As a good discipline, neuromarketing offers revolutionary methodologies for instantly penetrating consumers' minds without needing cognizant involvement and tends to provide opportunities for marketers and research scholars to inspect the neural mechanisms of consumer buying behavior [73–76]. However, neuroscience has exhibited a significant role in knowing consumer decision-making processes by discovering consumers' neural mechanisms and further aiding businesses to target potential consumers [77,78]. By commissioning neuromarketing tools in practice, companies could uncover numerous mysteries about what they do not understand about their consumers [79,80]. Visual art-based drawings on goods and

services or their wrapping materials significantly influence customer decision-making [81]. Moreover, neuromarketing tools support businesses in exploring consumers' minds based on physiological data, which are studied to identify and understand the diverse behaviors of consumers toward market offerings [82–86]. In a nutshell, it is time to go back to this prominent issue of whether the neuroscience approach to understanding the consumer mind and diverse behavior lives up to the hype and ensures the sustainable development of contemporary businesses [87].

3.1.6. Neuromarketing for Sustainable Business Development

Neuromarketing has recently earned substantial fame in the academic and commercial world and aided them in stimulating more significant insights into consumer behavior and sustainable consumption. In the mounting pattern of sustainable consumption, sustainable products, and their neuromarketing practices have played a decisive starring role [8]. Sustainable practices have become prevalent and are extensively employed in contemporary global business. In addition, they have been stimulating consumers' mindsets toward diverse marketing communications differently. Furthermore, innovative sustainable development trends such as green products, clean energy, green buildings, eco-innovations, etc., attract real estate customers toward sustainable consumption [88]. Therefore, neuromarketing tools can be a game-changer. However, neuromarketing approaches are still at an initial stage and must show their vast potential, and marketers will not halt their practices for sustainable business development [89]. Businesses attempt to unveil sustainable products and advertise via diverse marketing channels. Knowing consumer psychology and online behavior is critical to sustainable marketing, as consumer responses and decision-making no longer rely on traditional marketing. Thus, strategically implementing neuromarketing practices can open large doors for businesses and further deliver sustainable development [1].

*3.2. Context: Where Is the Research Happening?*

The aspect of 'Where' in the 4W framework [34] links with the Context (C) in the TCCM framework [35] and reports on the geographical and sector/industry contexts of the study.

Regarding the geographical contexts of the published articles, key countries where articles on neuromarketing have been published are Australia, Austria, Brazil, China, Denmark, Ecuador, France, Germany, Iran, Italy, Japan, Lithuania, and the Netherlands, as shown in Figure 2. Spain was identified as the country with the highest number of twelve published articles, followed by the United Kingdom with five published articles and the United States of America with three published articles on the neuromarketing domain.

The key sectors/industries identified in the published articles are tabulated in Table 2, which revealed "Advertising" as the top identified sector with 31.37% of the published articles, whereas, with 47.07% of the published articles, no sectors/industries were determined and stated as "Unknown."

**Table 2.** Sector/industry identified in the published articles.

| Sector/Industry | Frequency | Percentage | Exemplary Studies |
|---|---|---|---|
| Advertising | 16 | 31.37% | [8,12,63] |
| Beverage | 01 | 01.96% | [65] |
| Education | 01 | 01.96% | [90] |
| Food | 01 | 01.96% | [71] |
| Healthcare | 01 | 01.96% | [61] |
| Hospitality | 01 | 01.96% | [66] |
| Music | 01 | 01.96% | [64] |
| Real Estate | 01 | 01.96% | [85] |
| Retail | 03 | 05.88% | [91] |
| Toys and Games | 01 | 01.96% | [92] |
| Unknown | 24 | 47.07% | [56,60,93–95] |

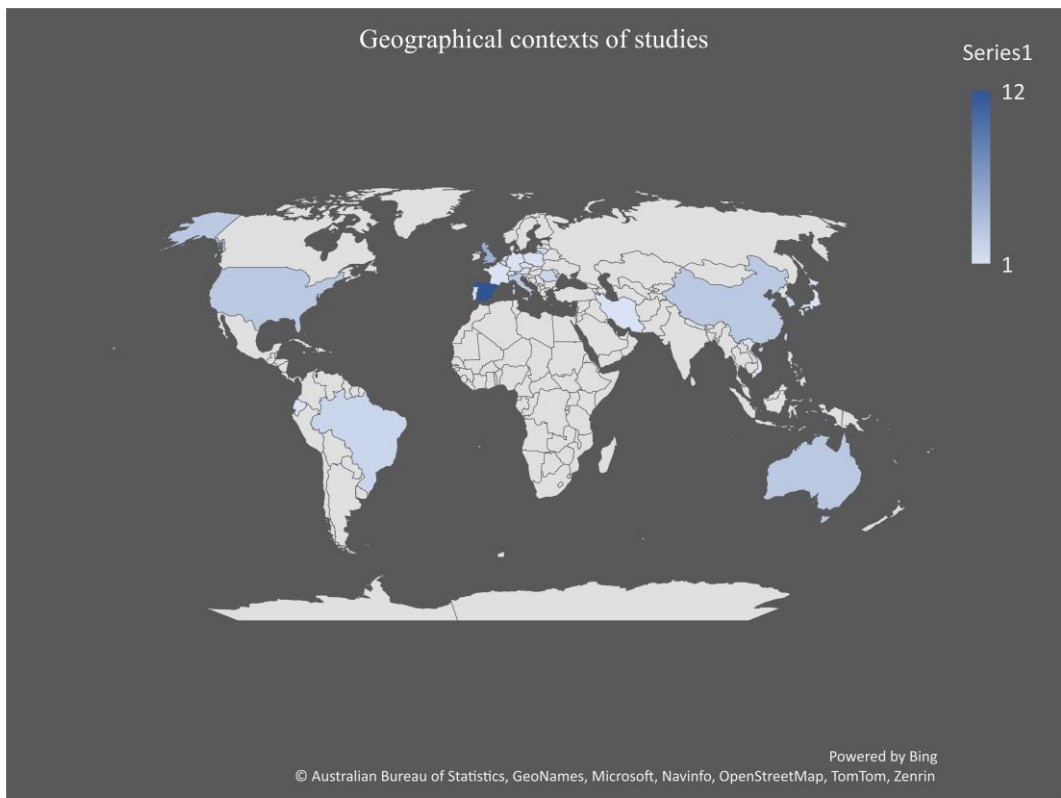

**Figure 2.** Geographical context of the published articles.

*3.3. Characteristics: Why Should Academicians, Practitioners, and Policymakers Know More about Neuromarketing?*

The aspect of 'Why' in the 4W framework [34] links with the Characteristics (C) in the TCCM framework [35] and reports on critical characteristics investigated in the study on neuromarketing. More precisely, we found the publication trends, key definitions of neuromarketing, essential neuromarketing tools, most cited articles, and key findings investigated in the 51 articles.

### 3.3.1. Characteristics: Publication Trends

The key results of the identified studies showed that no articles in the neuromarketing domain were published before 2010. As shown in Figure 3, 2021 had the highest number of 13 articles published in the SSCI category of the Web of Science index in the neuromarketing domain of research.

### 3.3.2. Characteristics: Neuromarketing Definitions, Neuromarketing Tools, and Author Keywords Occurrence

The key definitions of the term "neuromarketing" identified in the published articles are tabulated in Table 3. The essential neuromarketing tools identified in the published articles are tabulated in Table 4. The "Neuromarketing" keyword occurred most frequently in almost all the 51 articles, followed by the "consumer" and "marketing" keywords, as shown in Figure 4.

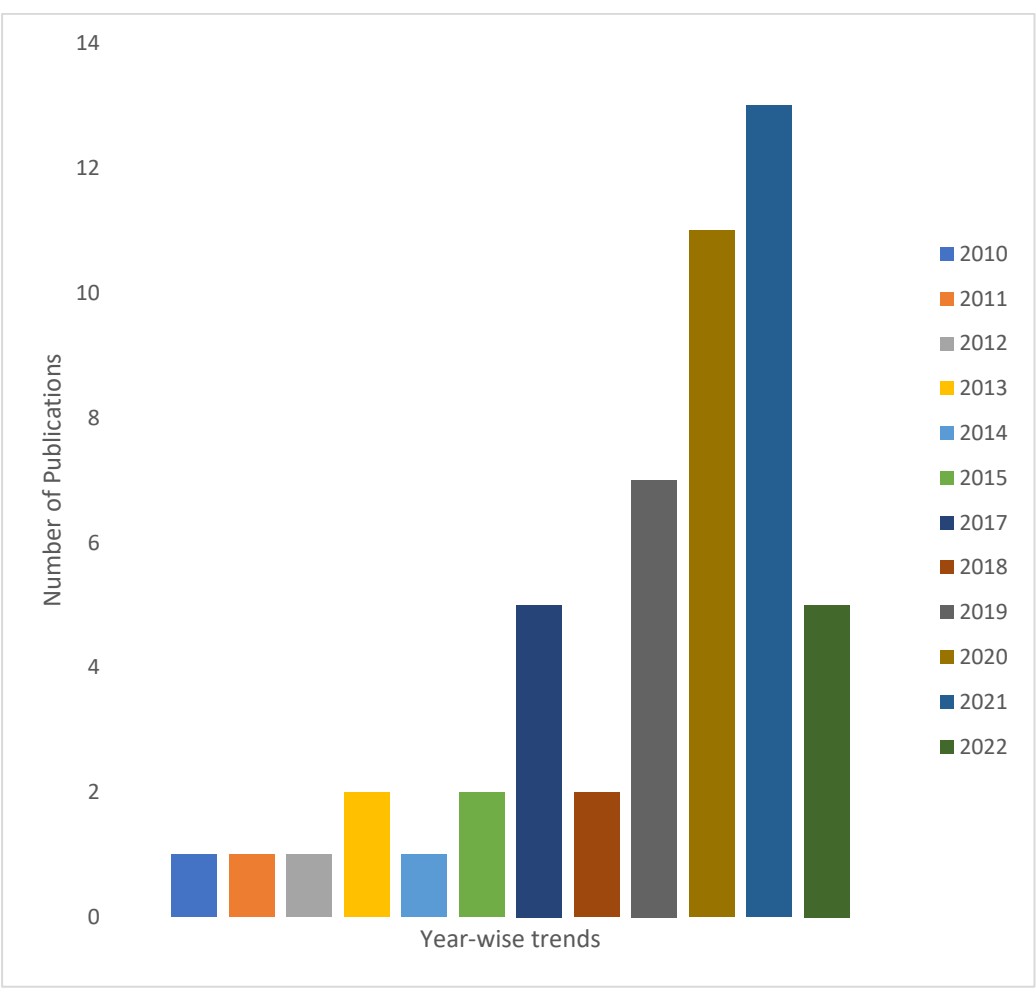

**Figure 3.** Year-wise trends in the published articles.

**Table 3.** The key definitions of neuromarketing are identified in the published articles.

| Author | Definition |
|---|---|
| [93] | "A marketing designed on the foundation of neuroscience study is one indicator of this innovative neuro-culture is termed as neuromarketing." |
| [96] | "The application of brain imaging and measurement techniques to identify consumer preferences is known as neuromarketing." |
| [97] | "Neuromarketing is using neuroscientific methods to analyze humans vis-à-vis market stimuli." |
| [69] | "Neuromarketing is a field of study that blends the conceptual understanding of behavioral psychology, consumer neuroscience, and economics." |
| [98] | "Neuromarketing is the usage of neurophysiological instruments to measure and quantitatively comprehend behaviors of humans relating to marketing product development." |
| [99] | "Neuromarketing is the field that attempts to apply the practices and understandings from consumer neuroscience in business applications and applied research." |

**Table 4.** Key neuromarketing tools identified in the published articles.

| Author | Neuromarketing Tools |
|:---:|:---|
| [90] | Functional magnetic resonance imaging (fMRI) |
| [94] | Functional magnetic resonance imaging (fMRI) and electroencephalography (EEG) |
| [60] | Functional magnetic resonance imaging (fMRI), electroencephalography (EEG), and peripheral biometric metrics |
| [92] | Electrodermal activity (EDA) and galvanic skin response (GSR) |
| [61] | Electroencephalography (EEG), functional magnetic resonance imaging (fMRI), galvanic skin response (GSR), heart rate (HR), facial coding, and eye tracking |
| [100] | Facial coding |
| [101] | Functional near-infrared spectroscopy (fNIRS) |
| [91] | Electromyography (EMG) |
| [102] | Facial coding, GSR, and eye tracking |

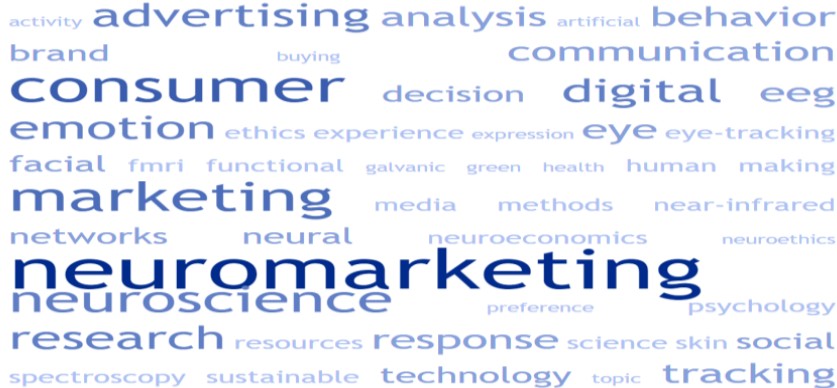

**Figure 4.** Word cloud of the most frequently occurring keywords.

### 3.3.3. Characteristics: Key Observations

A summary table of 51 eligible articles about neuromarketing research was created to present the identified key observations investigated in the published articles tabulated in Appendix A. Researchers, academics, and scholars might consider referring to the summary table to determine the nascent themes for future research directions and mature thematic areas as well as precise topics to conduct a study. They might employ limited utilized methodologies and limited explored geographical and sector/industry contexts to conduct neuromarketing research. Moreover, practitioners and policymakers who want to understand better the "neuromarketing" phenomenon might consider referring to the summary table to reach consumers more quickly and operatively by knowing more about this phenomenon.

### 3.3.4. Characteristics: Most Cited Articles

The most cited article was by Morin [73], with a total of 213 citations, followed by Fisher et al. [93], with a total of 83 citations; Javor et al. [68], with a total of 69 citations; Vecchiato et al. [82], with a total of 62 citations; and Lee et al. [55], with a total of 44 citations, as shown in Table 5. However, based on the average citation score, the rank position may vary for a few articles in this table.

**Table 5.** Top 5 articles (ranked by total citation score up to 31 December 2022).

| Rank Position | Author | Journal | Total Citation | Average Citation |
|---|---|---|---|---|
| 01 | [73] | *Society* | 213 | 16.38 |
| 02 | [93] | *Harvard Review of Psychiatry* | 83 | 05.93 |
| 03 | [68] | *BMC Neurology* | 69 | 06.27 |
| 04 | [82] | *Computational and Mathematical* | 62 | 06.20 |
| 05 | [55] | *Methods in Medicine* | 44 | 07.33 |
| | | *European Journal of Marketing* | | |

*3.4. Methodology: HoW Was the Research Conducted?*

The aspect of 'HoW' in the 4W framework [17] links with the Methodology (M) in the TCCM framework [33] and reports on critical methodologies investigated in the study on neuromarketing. The key methodologies identified in the published articles are tabulated in Table 6.

**Table 6.** Methodologies investigated in the published articles.

| Methodology | Frequency | Percentage | Exemplary Studies |
|---|---|---|---|
| Conceptual | 14 | 27.45% | [60,93–95] |
| Qualitative research | 27 | 52.94% | [62,63,92] |
| Quantitative research | 07 | 13.73% | [7,12,64] |
| Mixed research study | 03 | 05.88% | [69,97] |

It was observed that 52.94% of the published articles were based on a qualitative research approach, 14% of the published articles were based on a conceptual research approach, 13.73% of the published articles were based on a quantitative research approach, and 05.88% of the published articles were based on a mixed research study approach. Hence, it was discovered that there is also a lack of literature, particularly on mixed research study-based articles, which may have supported theme development vis-à-vis the neuromarketing domain of research. Also, review articles based on a conceptual research approach are the prerequisite of the hour.

**4. Limitations, Future Research Directions, and Implications**

*4.1. Limitations*

Nothing is persistently perfect, and the rule even now can stand as right in the context of this study, which has several limitations, even though this study has attempted to highlight the critical contexts where neuromarketing is in practice. Still, this study has several limitations that propose future research opportunities. The first limitation of this study was that it is based on obtaining eligible articles from the Web of Science database only. In addition, the second limitation of this study was the research protocol. Only those articles indexed in the SSCI category of the Web of Science index were selected, articles written in English were selected, and the document type considered was "Articles & Review articles". The following limitation was the final sample size, which comprised 51 articles published up to 31 December 2022. For exact search results, the study used the search words "neuromarketing" or "neuro-marketing," which was the fourth limitation of this study. Finally, the fifth limitation of this study was the methodology. The analysis is conceptual and is based on the SLR approach. However, the aim was to deepen and heighten theoretical acumen about the neuromarketing phenomenon, and hopefully, this study will accomplish this aim.

*4.2. Future Research Directions*

Regardless of the multi-disciplinary nature of the "neuromarketing" phenomenon, which has occasioned diverse systematic production in the extant literature, the present

study has demarcated its prominent positions of study from a holistic approach based on the SLR approach. This field of study is still nascent, so there is a need to set the tone for future research directions. For this purpose, future research questions (FRQs) will be proposed. The term FRQ was proposed in the work of [44]. The blend of the 4W and TCCM frameworks has proposed future research directions in critical aspects. Based on the critical aspects shown in Table 7, the future research directions are more precisely offered and presented in FRQs.

**Table 7.** Proposed future research questions (FRQs) for future research directions.

| Aspect (s) | Future Research Questions (FRQs) |
|---|---|
| *'What'—'Theme'* | **FRQ1:** What substantial themes are identified in neuromarketing research?<br>**FRQ2:** What are the critical behavioral and psychological and behavioral aspects of neuromarketing in arguing with contemporary research fields like consumer behavior, the metaverse, digital engagement, etc.?<br>**FRQ3:** What are the necessary marketing stimuli forms presently applied in neuromarketing? |
| *'Where'—'Context'* | **FRQ1:** Where are the neuromarketing research studies happening in diverse geographical contexts?<br>**FRQ2:** Which are the key sectors where neuromarketing research studies are happening?<br>**FRQ3:** What are the contemporary areas that have emerged in the contemporary period in neuromarketing research?<br>**FRQ4:** In what way does neuromarketing create an impact on marketing and brand building?<br>**FRQ5:** In what way does the neuromarketing phenomenon impact the digital marketing field?<br>**FRQ6:** What are the critical consequences of neuromarketing on consumer psychology?<br>**FRQ7:** What are the critical consequences of neuromarketing on the consumer buying decision process?<br>**FRQ8:** What is the notable application of neuromarketing at the local and global levels in the E-commerce industry?<br>**FRQ9:** What is the notable application of neuromarketing at the local and global levels in the education sector?<br>**FRQ10:** What is the notable application of neuromarketing at the local and global levels in the fashion industry?<br>**FRQ11:** What is the notable application of neuromarketing at the local and global levels in business?<br>**FRQ12:** In what way are neuromarketing tools used in practice in knowing consumer behavior?<br>**FRQ13:** What are the publication trends in neuromarketing research? |
| *'Why'—'Characteristics'* | **FRQ1:** What are the key constructs of neuromarketing research?<br>**FRQ2:** What is the history of neuromarketing research?<br>**FRQ3:** What are the fundamental definitions of neuromarketing?<br>**FRQ4:** Which article is cited the most times in neuromarketing research?<br>**FRQ5:** Who are the key authors in neuromarketing research?<br>**FRQ6:** What are neuromarketing research studies' possible antecedents, decisions, and outcomes?<br>**FRQ7:** What essential role does neuromarketing play in drafting marketing strategies in business?<br>**FRQ8:** In what way does neuromarketing affect consumer attitudes and habits?<br>**FRQ9:** What are the critical challenges for businesses in adopting neuromarketing tools? |
| *'HoW'—'Methodology'* | **FRQ1:** What are the critical methodologies employed in neuromarketing research?<br>**FRQ2:** Which is the top-most employed methodology in neuromarketing research among researchers from developing economies?<br>**FRQ3:** Which is the top-most employed methodology in neuromarketing research among researchers from developed economies?<br>**FRQ4:** Which is the top-most employed methodology in neuromarketing research among researchers from under-developed economies? |

*4.3. Implications*

4.3.1. Academic Implications

The neuromarketing research era in the recent decade has shown that this area is a promising field of study for the academic world in the long run. These perceptions necessitate immediate attention and action to widen the scope from both ends and to make neuromarketing an academic and practical concept. The neuromarketing phenomenon has commonly been employed to characterize customers' preferences accurately. Even though

businesses are disinclined to divulge their findings on enhancing marketing strategies, the best place to begin for a new academic is to become acquainted with the underlying strengths and weaknesses of utilizing neuromarketing as a balancing technique to replace an autonomous approach. Furthermore, the neuromarketing insights contributed by this study can assist academics in developing marketing phenomena explanations in a better way [79]. Researchers new to the field should also utilize the existing literature to find areas regarding the substantial implications of neuromarketing to find exciting topics since this kind of reminiscent theoretical work is plentiful. Future research investigations can be conducted by referring to this work by selecting any future research questions proposed and presented in Table 7 to heighten the acumen on the neuromarketing phenomenon. Secondly, academic content might be developed based on this study to assist learners in better understanding this phenomenon. Finally, researchers and scholars planning to conduct further studies on neuromarketing might consider referring to this study to identify research gaps and may refer to categorize research themes [95].

### 4.3.2. Practical Implications

This work may broaden the understanding of businesses and marketers on how neuromarketing is employed in practice and what essential neuromarketing tools might be utilized. As for practical implications, our research acknowledges the prominence of neuromarketing for marketing practitioners. By researching the sentient and intuitive reactions of brains, contemporary businesses can alter their imminent business and marketing strategies to meet their organizational goals [103]. Marketers might consider applying this study's theoretical understanding to develop branding strategies for creating effective advertisements [14].

### 5. Conclusions

This study presents an inherent theoretical backdrop vis-à-vis the neuromarketing phenomenon by systematically exploring 51 articles on neuromarketing from the Web of Science database. The SLR and PRISMA approaches were employed to study pertinent and integrated expertise in this research domain. Furthermore, the 4W and TCCM frameworks were used to present the key results, categorizing 'What' with key 'Themes' investigated, 'Where' with key 'Contexts' of studies in diverse geographical and sectors, 'Why' with key 'Characteristics' investigated stating publication trends and key observations, and 'HoW' with 'Methodology' investigated which were employed to conduct studies. Neuromarketing study is notable for its exploration of evolution, definitions, tools, practices, social value, influence on consumer behavior, and potential for sustainable business development within the discipline. Neuromarketing science and practice are improved by the findings of this study, which provides a historical perspective on the term's evolution, tracing its origins from James Vicary's experiments to formal recognition in the early 2000s. By examining the field's history, readers can better understand how it developed. This article stresses the importance of collaboration and wisdom integration between neuroscience, psychology, marketing, engineering, and economics, underscoring that neuromarketing is an interdisciplinary field of research.

With the dawn of digital marketing and E-commerce, it is more challenging for businesses and marketers to know consumers' diverse and disruptive behavior across the globe. Recent catastrophic events such as the COVID-19 pandemic and financial catastrophe contributed to this disruption in consumer attitudes and overall behavior [104,105]. Thus, knowing what is going on in a consumer's mind is crucial. Further, the study provides a comprehensive definition of neuromarketing as the study of consumers' sensory responses to marketing communications, as well as highlighting the importance of understanding how consumers' instinctual minds react to marketing stimuli, which can facilitate the formulation of more effective marketing strategies. The study discusses the various tools and technologies employed in the neuromarketing discipline, such as fMRI, EEG, eye tracking, and facial expression assessment, and underlines these tools' evolution and rel-

evancy in researching consumer behavior. It is a call to action for businesses to study neuromarketing tools and their prospect to attain insights that conventional marketing approaches may not provide. Here, neuromarketing is essential for businesses as it is instrumental in understanding consumer behaviors [103]. Moreover, it pinpoints that while neuromarketing has attained business and academic concentration, its practical application demands improvement. It presents the critical matter of upholding the social value of neuromarketing study and fosters ethical deliberations in neuromarketing practices.

The study also spotlights the importance of neuromarketing in comprehending consumer decision-making processes and underlines the function of neuroscience in discovering consumers' consciousness and how this understanding can be employed by businesses to segment and target probable consumers. In topical years, the neuromarketing literature has expanded extraordinarily [106]. Even though this phenomenon has triggered a distinct body of knowledge, there is a scarcity of research categorizing the prevailing literature [30,106–108]. A substantial contribution of the study is its focus on the function of neuromarketing in the sustainable development of businesses across the globe. The study underscores how neuromarketing can assist in fostering sustainable products and business practices by understanding consumer behavior and psychology. The study advances businesses in strategically applying neuromarketing practices, identifying that consumer reactions and decision-making have matured in the digital era. Neuromarketing is a comprehensive discipline that demands the use of neuroscientific tools and techniques to discover consumers' known and unknown behaviors, particularly in E-commerce [109,110]. This signifies that neuromarketing can unlock doorways for businesses and contribute to sustainable business development. This neuromarketing study conveys valuable insights to scholars, researchers, and marketers by offering comprehensive future research perspectives and theoretical evolution for augmenting their overall understanding of the neuromarketing discipline.

**Author Contributions:** Conceptualization, P.S. and I.A.; methodology, P.S.; software, P.S.; validation, P.S., I.A. and L.K.; formal analysis, P.S.; investigation, P.S., L.K. and I.A.; resources, P.S., L.K. and I.A.; data curation, P.S. and L.K.; writing—P.S.; visualization, P.S. and I.A.; supervision, I.A. and L.K. All authors have read and agreed to the published version of the manuscript.

**Funding:** This research received no external funding.

**Institutional Review Board Statement:** Not applicable.

**Informed Consent Statement:** Not applicable.

**Data Availability Statement:** All data are included within the article.

**Acknowledgments:** The authors thank the journal's editor and anonymous reviewers for their valuable feedback during the review process.

**Conflicts of Interest:** The authors declare no conflict of interest.

## Appendix A

**Table A1.** Summary Table of 51 Eligible Articles.

| S. No. | Authors | Source Title | Year | Country | Sector/Industry | Methodology | Themes | Neuromarketing Tools | Key Contributions | Total Citations |
|---|---|---|---|---|---|---|---|---|---|---|
| 1 | Lee, N; Brandes, L; Chamberlain, L; Senior, C [81] | *Journal of Marketing Management* | 2017 | United Kingdom | Unknown | Conceptual | Neuromarketing tools | Functional magnetic resonance imaging (fMRI) and electroencephalography (EEG) | Proposed a schematic design for the fundamental process of a conventional neuromarketing research | 21 |
| 2 | Fisher, CE; Chin, L; Klitzman, R [80] | *Harvard Review of Psychiatry* | 2010 | United States of America | Unknown | Conceptual | Neuromarketing definitions | Functional magnetic resonance imaging (fMRI) and electroencephalography (EEG) | Reviewed neuromarketing history | 83 |
| 3 | Gonzalez-Morales, A; Mitrovic, J; Garcia, RC [47] | *European Research on Management and Business Economics* | 2020 | Spain | Unknown | Conceptual | Neuromarketing tools | Functional magnetic resonance imaging (fMRI), electroencephalography (EEG), and peripheral biometric metrics | Observed neuromarketing tools' applications in the ecological branding strategy | 03 |
| 4 | Zhu, ZR; Jin, YQ; Su, YS; Jia, K; Lin, CL; Liu, XX [43] | *Frontiers in Psychology* | 2022 | China | Unknown | Conceptual | Evolution of neuromarketing | Eye tracking, fMRI, and EEG/ERPs | Offered an overview of key trends in neuromarketing | 02 |
| 5 | Brenninkmeijer, J; Schneider, T; Woolgar, S [49] | *Science Technology, & Human Values* | 2020 | The Netherlands | Unknown | Qualitative research | Neuromarketing in practice | Functional magnetic resonance imaging (fMRI) | Offered the outcomes of ethnographic research on neuromarketing practice | 07 |
| 6 | Ahmed, RR; Streimikiene, D; Channar, ZA; Soomro, HA; Streimikis, J; Kyriakopoulos, GL [12] | *Sustainability* | 2022 | United States of America | Advertising Industry | Quantitative research | Evolution of neuromarketing | Functional magnetic resonance imaging (fMRI), eye tracking, electroencephalography (EEG), and steady-state probe topography (SSPT) | Recommended a neural network as an option compared to conventional neuromarketing tools | 01 |
| 7 | Cardoso, L; Chen, MM; Araujo, A; de Almeida, GGF; Dias, F; Moutinho, L [82] | *Behavioral sciences* | 2022 | Portugal | Unknown | Conceptual | Evolution of neuromarketing | Functional magnetic resonance imaging (fMRI) and electroencephalography (EEG) | Offered an improved perception of neuromarketing studies | 03 |

**Table A1.** *Cont.*

| S. No. | Authors | Source Title | Year | Country | Sector/Industry | Methodology | Themes | Neuromarketing Tools | Key Contributions | Total Citations |
|---|---|---|---|---|---|---|---|---|---|---|
| 8 | Nilashi, M; Yadegaridehkordi, E; Samad, S; Mardani, A; Ahani, A; Aljojo, N; Razali, NS; Tajuddin, T [8] | *Symmetry—Basel* | 2020 | Vietnam | Advertising Industry | Quantitative research | Neuromarketing for sustainable business development | Facial recognition/facial coding system (FACS), heart rate (HR), electroencephalography (EEG), galvanic skin response (GSR), eye tracking (ET), positron emission tomography (PET), magnetoencephalography (MEG), and fMRI. | Disclosed critical causes for neuromarketing applications in business settings | 10 |
| 9 | Constantinescu, M; Orindaru, A; Pachitanu, A; Rosca, L; Caescu, SC; Orzan, MC [7] | *Sustainability* | 2019 | Romania | Unknown | Quantitative research | Neuromarketing in practice | EEG, voice recognition, eye tracking, fMRI, and face coding | Discussed neuromarketing research from a social media perspective | 10 |
| 10 | Lee, N; Chamberlain, L; Brandes, L [42] | *European Journal of Marketing* | 2018 | United Kingdom | Unknown | Conceptual | Evolution of neuromarketing | Functional Magnetic Resonance Imaging (fMRI) | Found mounting interest in the neuromarketing field | 44 |
| 11 | Levallois, C; Smidts, A; Wouters, P [35] | *Business History* | 2021 | France | Unknown | Conceptual | Evolution of neuromarketing | Functional magnetic resonance imaging (fMRI), and electroencephalography (EEG) | Talked about the emergence of neuromarketing | 05 |
| 12 | Banos-Gonzalez, M; Baraybar-Fernandez, A; Rajas-Fernandez, M [50] | *Frontiers in Psychology* | 2020 | Spain | Advertising Industry | Qualitative research | Neuromarketing in practice | Electroencephalogram (EEG) or magnetic resonance imaging (MRI) | Discussed neuromarketing in a scientific study regarding consumer behavior | 05 |
| 13 | Bradfield, OM [54] | *Neuroethics* | 2021 | Australia | Unknown | Conceptual | Social value of neuromarketing research | Neuroimaging research incidental findings (NRIFs) | Explained how businesses are employing neuromarketing in practice | 02 |

**Table A1.** *Cont.*

| S. No. | Authors | Source Title | Year | Country | Sector/Industry | Methodology | Themes | Neuromarketing Tools | Key Contributions | Total Citations |
|---|---|---|---|---|---|---|---|---|---|---|
| 14 | Schneider, T; Woolgar, S [83] | *Consumption Markets & Culture* | 2012 | United Kingdom | Advertising Industry | Conceptual | Neuromarketing and consumer behavior | Functional magnetic resonance imaging (fMRI), electroencephalography (EEG), magnetoencephalography (MEG), steady-state topography (SST), and positron emission tomography (PET) | Discussed diverse neuromarketing tools | 39 |
| 15 | Morin, C [60] | *Society* | 2011 | United States of America | Advertising Industry | Conceptual | Neuromarketing and consumer behavior | Electroencephalography (EEG), magnetoencephalography (MEG), and functional magnetic resonance imaging (fMRI) | Discovered that neuromarketing is a promising area of study | 213 |
| 16 | Schneider, T; Woolgar, S [66] | *Biosocieties* | 2015 | United Kingdom | Unknown | Conceptual | Neuromarketing definitions | N.S. | Examined the emergence of neuromarketing | 12 |
| 17 | Crespo-Pereira, V; Garcia-Soidan, P; Martinez-Fernandez, VA [84] | *Profesional de la informació n* | 2019 | Ecuador | Advertising Industry | Mixed research study | Neuromarketing definitions | EEG and eye tracking | Explored the influence of neuromarketing on TV channels | 12 |
| 18 | Spence, C [74] | *Organizational Research Methods* | 2019 | United Kingdom | Unknown | Conceptual | Neuromarketing and consumer behavior | Functional magnetic resonance imaging (fMRI), electroencephalography (EEG), magnetoencephalography—MEG, and positron emission tomography—PET | Concentrated on three crucial sections namely, neurogastronomy, neuroergonomics, and neuromarketing | 36 |
| 19 | Zito, M; Fici, A; Bilucaglia, M; Ambrogetti, FS; Russo, V [73] | *Frontiers in Psychology* | 2021 | Italy | Unknown | Qualitative research | Neuromarketing and consumer behavior | Electroencephalography (EEG), skin conductance (SC), and eye+ tracking | Studied neuromarketing techniques | 05 |
| 20 | Javor, A; Koller, M; Lee, N; Chamberlain, L; Ransmayr, G [55] | *BMC Neurology* | 2013 | Austria | Unknown | Qualitative research | Social value of neuromarketing research | fMRI, EEG and galvanic skin response | Discussed the application of neuroscientific techniques | 69 |

**Table A1.** *Cont.*

| S. No. | Authors | Source Title | Year | Country | Sector/Industry | Methodology | Themes | Neuromarketing Tools | Key Contributions | Total Citations |
|---|---|---|---|---|---|---|---|---|---|---|
| 21 | Duan, L; Ai, H; Yang, LL; Xu, LL; Xu, PF [62] | *Frontiers in Psychology* | 2021 | China | Advertising Industry | Qualitative research | Neuromarketing and consumer behavior | Functional near-infrared spectroscopy (fNIRS) | Debated the usage of neuroscientific techniques | 05 |
| 22 | Nunez-Cansado, M; Lopez, AL; Dominguez, DC [56] | *Frontiers in Psychology* | 2020 | Spain | Unknown | Mixed research study | Neuromarketing definitions | EMG, ECG/EKG, or fMRI, and facial coding | Examined the application of neuroscientific tools in practice | 02 |
| 23 | Sanchez-Nunez, P; Cobo, MJ; Vaccaro, G; Pelaez, JI; Herrera-Viedma, E [86] | *Brain Sciences* | 2021 | Spain | Unknown | Conceptual | Neuromarketing definitions | Functional magnetic resonance Imaging (fMRI) | Discovered neuromarketing as a broad research field | 02 |
| 24 | Mauri, M; Rancati, G; Gaggioli, A; Riva, G [87] | *Frontiers in Psychology* | 2021 | Italy | Unknown | Qualitative research | Neuromarketing tools | Facial coding | Explored neuromarketing techniques | 00 |
| 25 | Leeuwis, N; Pistone, D; Flick, N; van Bommel, T [51] | *Frontiers in Psychology* | 2021 | The Netherlands | Music industry | Quantitative research | Neuromarketing in practice | Functional magnetic resonance imaging (fMRI), and electroencephalography (EEG) | Examined the application of neuroscientific tools in practice | 01 |
| 26 | Varan, D; Lang, A; Barwise, P; Weber, R; Bellman, S [76] | *Journal of Advertising Research* | 2015 | Australia | Advertising Industry | Qualitative research | Neuromarketing for sustainable business development | Functional magnetic Resonance imaging (fMRI), and electroencephalography (EEG) | Talked about neuromarketing methods | 40 |
| 27 | Garczarek-Bak, U; Szymkowiak, A; Gaczek, P; Disterheft, A [63] | *Journal of Brand Management* | 2021 | Poland | Advertising Industry | Qualitative research | Neuromarketing and consumer behavior | Electroencephalography (EEG), electrodermal activity (EDA), and eye-tracking (ET) | Examined the application of neuromarketing tools in practice | 07 |
| 28 | Nunez-Gomez, P; Alvarez-Ruiz, A; Ortega-Mohedano, F; Alvarez-Flores, EP [97] | *Frontiers in Psychology* | 2020 | Spain | Advertising Industry | Qualitative research | Social value of neuromarketing research | Functional magnetic resonance imaging (fMRI), and electroencephalography (EEG) | Talked about neuromarketing tools | 01 |
| 29 | Lindell, AK; Kidd, E [77] | *Mind, Brain, and Education* | 2013 | Australia | Education Industry | Qualitative research | Neuromarketing tools | Functional magnetic resonance imaging (fMRI) | Explained neuromarketing in practice | 12 |

**Table A1.** *Cont.*

| S. No. | Authors | Source Title | Year | Country | Sector/Industry | Methodology | Themes | Neuromarketing Tools | Key Contributions | Total Citations |
|---|---|---|---|---|---|---|---|---|---|---|
| 30 | Zhao, MN; Wang, J; Zhang, H; Zhao, G [65] | *Sustainability* | 2019 | China | Retail industry | Quantitative research | Neuromarketing and consumer behavior | Electroencephalography (EEG) | Described the psychological actions and neurological actions of the human brain | 05 |
| 31 | Micu, A; Capatina, A; Micu, AE; Geru, M; Aivaz, KA; Muntean, MC [64] | *Economic Computation and Economic Cybernetics Studies and Research* | 2021 | Romania | Unknown | Conceptual | Neuromarketing and consumer behavior | Electroencephalography (EEG), magnetoencephalography (MEG), functional magnetic resonance imaging (fMRI), galvanic skin response (GSR), electrocardiogram (ECG), facial coding, and eye tracking | Offered comprehensive mapping of neuromarketing applications in a social media study | 00 |
| 32 | Kurahashi, C; Misawa, T; Yamashita, K [72] | *Sensors and Materials* | 2018 | Japan | Advertising Industry | Qualitative research | Neuromarketing and consumer behavior | Near-infrared spectroscopy (NIRS) and EEG | Investigated the consequences of advertisement on consumers | 03 |
| 33 | Vecchiato, G; Maglione, AG; Cherubino, P; Wasikowska, B; Wawrzyniak, A; Latuszynska, A; Latuszynska, M; Nermend, K; Graziani, I; Leucci, MR; Trettel, A; Babiloni, F [69] | *Computational and Mathematical Methods in Medicine* | 2014 | Italy | Advertising Industry | Qualitative research | Neuromarketing and consumer behavior | Electroencephalographic (EEG), galvanic skin response (GSR), and heart rate (HR) | Examined consumers' response to advertisements from a neuroscientific viewpoint | 62 |
| 34 | Ramsoy, TZ; Jacobsen, C; Friis-Olivarius, M; Bagdziunaite, D; Skov, M [71] | *Journal of Neuroscience Psychology and Economics* | 2017 | Denmark | Unknown | Qualitative research | Neuromarketing and consumer behavior | Galvanic skin response (GSR) and eye tracking | Discussed neuromarketing applications in businesses | 12 |
| 35 | Baraybar-Fernandez, A; Banos-Gonzalez, M; Barquero-Perez, O; Goya-Esteban, R; de-la-Morena-Gomez, A [70] | *Comunicar* | 2017 | Spain | Advertising Industry | Quantitative research | Neuromarketing and consumer behavior | Electrodermal activity (EDA), ECG, and HR | Studied consumers' emotions regarding advertising messages | 33 |

**Table A1.** *Cont.*

| S. No. | Authors | Source Title | Year | Country | Sector/Industry | Methodology | Themes | Neuromarketing Tools | Key Contributions | Total Citations |
|---|---|---|---|---|---|---|---|---|---|---|
| 36 | Guixeres, J; Bigne, E; Azofra, JMA; Raya, MA; Granero, AC; Hurtado, FF; Ornedo, VN [61] | *Frontiers in Psychology* | 2017 | Spain | Advertising Industry | Qualitative research | Neuromarketing and consumer behavior | Electroencephalography (EEG), functional magnetic resonance imaging (fMRI), HR, electrocardiogram (ECG), facial coding, and eye tracking | Researched the efficiency of the latest ADs on digital channels | 43 |
| 37 | Chiang, MC; Yen, CH; Chen, HL [1] | *Sustainability* | 2022 | Taiwan | Retail industry | Qualitative research | Neuromarketing for sustainable business development | Eye tracking, EEG, and face reading | Researched physiological measures | 00 |
| 38 | Hsu, LW; Chen, YJ [52] | *British Food Journal* | 2019 | Taiwan | Bevarage Industry | Qualitative research | Neuromarketing in practice | Electroencephalography (EEG) | Explored the grooming influence of music on participant wine tasting fondness | 11 |
| 39 | Boscolo, JC; Oliveira, JHC; Maheshwari, V; Giraldi, JDE [67] | *Marketing Intelligence & Planning* | 2020 | Brazil | Advertising Industry | Qualitative research | Neuromarketing and consumer behavior | Eye tracking and EEG | Examined attitudes towards various advertisement types | 04 |
| 40 | Kaklauskas, A; Ubarte, I; Kalibatas, D; Lill, I; Velykorusova, A; Volginas, P; Vinogradova, I; Milevicius, V; Vetloviene, I; Grubliauskas, R; Bubliene, R; Naumcik, A [75] | *Energies* | 2019 | Lithuania | Real Estate Industry | Qualitative research | Neuromarketing for sustainable business development | Heart rate and face reading | Talked about neuromarketing method applications in real estate | 03 |
| 41 | Akbarialiabad, H; Bastani, B; Taghrir, MH; Paydar, S; Ghahramani, N; Kumar, M [48] | *Frontiers in Psychiatry* | 2021 | Iran | Healthcare Industry | Qualitative research | Neuromarketing tools | Electroencephalography (EEG), functional magnetic resonance imaging (fMRI), galvanic skin response (GSR), heart rate, facial coding, and eye tracking | Investigated voter's neuronal response regarding casting votes and decision=-making | 02 |
| 42 | Juarez, D; Tur-Vines, V; Mengual, A [79] | *Frontiers in Psychology* | 2020 | Spain | Toys & Games Industry | Qualitative research | Neuromarketing tools | Electrodermal activity (EDA) and galvanic skin response (GSR) | Discussed ways of assessing brain action | 09 |

**Table A1.** *Cont.*

| S. No. | Authors | Source Title | Year | Country | Sector/Industry | Methodology | Themes | Neuromarketing Tools | Key Contributions | Total Citations |
|---|---|---|---|---|---|---|---|---|---|---|
| 43 | Grigaliunaite, V; Pileliene, L [57] | *Oeconomia copernicana* | 2017 | Lithuania | Unknown | Qualitative research | Social value of neuromarketing research | N.S | Studied attitudes towards smoking and behavioral outcomes using images promoting no smoking | 03 |
| 44 | Mengual-Recuerda, A; Tur-Vines, V; Juarez-Varon, D [53] | *Frontiers in Psychology* | 2020 | Spain | Hospitality Industry | Quantitative research | Neuromarketing in practice | Eye tracking, galvanic skin response (GSR), and electroencephalography (EEG) | Specified emotion and neuromarketing connection | 10 |
| 45 | Barquero-Perez, O; Camara-Vazquez, MA; Vadillo-Valderrama, A; Goya-Esteban, R [85] | *Frontiers in Psychology* | 2020 | Spain | Unknown | Qualitative research | Neuromarketing definitions | Heart rate, fMRI, and electrodermal activity (EDA) | Discussed the influence of advertisements on prospective consumers | 04 |
| 46 | Kim, Y; Park, K; Kim, Y; Yang, W; Han, D; Kim, WS [68] | *Frontiers in Psychology* | 2020 | South Korea | Unknown | Qualitative research | Neuromarketing and consumer behavior | Functional magnetic resonance imaging (fMRI), and electroencephalography (EEG) | Discussed the application of visual-art-based designs and the influence on consumers' buying decision-making process | 07 |
| 47 | Qing, KQ; Huang, RS; Hong, KS [88] | *Frontiers in Human Neuroscience* | 2021 | South Korea | Advertising Industry | Qualitative research | Neuromarketing tools | Functional near-infrared spectroscopy (fNIRS) | Decrypted consumers' fondness levels | 06 |
| 48 | Levrini, GRD; dos Santos, MJ [78] | *Behavioral Sciences* | 2021 | Brazil | Retail industry | Qualitative research | Neuromarketing tools | Electromyography (EMG) | Talked about sensory reactions and the relationship with consumers' inclination | 06 |
| 49 | Rua-Hidalgo, I; Galmes-Cerezo, M; Cristofol-Rodriguez, C; Aliagas, I [89] | *Behavioral Sciences* | 2021 | Spain | Unknown | Mixed research study | Neuromarketing tools | Facial coding, GSR, and eye tracking | Talked about neuroscience study methods | 03 |
| 50 | Madipakkam, AR; Bellucci, G; Rothkirch, M; Park, SQ [58] | *Scientific Reports* | 2019 | Germany | Food Industry | Qualitative research | Social value of neuromarketing research | Eye-tracking | Researched the impact of gaze of one customer on another while selecting a product | 07 |
| 51 | Rodriguez-Fuertes, A; Alard-Josemaria, J; Sandubete, JE [59] | *Frontiers in Psychology* | 2022 | Spain | Unknown | Qualitative research | Social value of neuromarketing research | Emotionally congruent facial responses (ECFR) | Discussed neuromarketing tools | 00 |

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
