# Peer review of "What Do You Need to Know? A Systematic Review and Research Agenda on Neuromarketing Discipline"

_jtaer, doi:10.3390/jtaer18040101_

Round 1

Reviewer 1 Report

Comments and Suggestions for Authors

The paper is well-organized and well-written! It respects the systematic review, structure, and flow.

I recommend, prior acceptance for publication:

1. To format Table 1 according to the journal requirements.

2. Improve Figure 5. Now it looks like a draft.

Comments on the Quality of English Language

Minor editing of English language required

Reviewer 2 Report

Comments and Suggestions for Authors

Thanks fot this opportunity to review and contribute to this article. It catched up my attention and interest.

The title. Less is more. “Research Agenda on Neuromarketing discipline” maybe…

When citing figure 1 and figure 2 in texts something happens with distancing. Recheck.

After figure 2, at page 10, “Error!360 Reference source not found.”

Some of your titles along the Manuscript, should also be readapt. By instance: “4.3 Characteristics: Why should academicians, practitioners, and policymakers know more about neuromarketing”.

Figure 3 is too complex. Let only the years above the bars.

Good Luck!

Comments on the Quality of English Language

Proofreading before new submission is recommended.

Reviewer 3 Report

Comments and Suggestions for Authors

Dear Author(s),

 Thank you for your paper. 

These are very minor revisions suggested, as I believe this paper would be a great fit for the journal:

Line 21 ´The key results include six key themes´. Please be so kind as to List in abstract all six key themes

Line 20 and line 23. No need to repeat the number of articles analyzed

Better link with electronic commerce research in the abstract and all the paper. It appears only in the conclusion. Could be relevant to have link between conclusions and the rest of the work.

Line 87-96. Better link with electronic commerce research

Explain the theoretical model mentioned in the line 22 and what is new in Line 451, competing with other studies. What makes believe that it is complete.

Editing issues. For example, line 360, 403 e etc

Explain what is the importance of this paper and what new it brings to the science and practice specifically.  Section 6.3 is very generic. Tangible conclusions are missing.

Line 168. Explain how papers were excluded, detailed description of the criteria is missing. Not very clear why article is considered eligible.

Literature review could be stronger with references globally recognized in the field. For example, António Damásio and Philip Kotler.

Thank you again for your work and I wish you all the best in your future research!

Reviewer 4 Report

Comments and Suggestions for Authors

This is an interesting research topic, aiming to perform the systematic analysis of the current state of neuromarketing research.

Abstract. The aim of the article is not indicated. Also, scientific value and originality must be emphasized. Also, as systematic reviews in the field of neuromarketing are quite popular, the novelty and originality must be emphasized.

Introduction.  The history of neuromarketing and its essence are described. However, the current state of the research in the domain is not demonstrated, and the findings obtained by prominent authors are not presented. Therefore, the gap in the literature which the authors intend to close is not revealed. The aim or purpose of this systematic review is not clear. The research questions and objectives are provided; however, they must correspond to each other.

I recommend enhancing the Introduction part by incorporating more recent references to bolster its quality. Four references that have the potential to enrich your introduction by introducing captivating topics and engaging insights, as follows:

https://doi.org/10.3390/su15054603

https://doi.org/10.1007/s12144-023-04812-w

https://doi.org/10.3991/ijoe.v18i08.31959

https://doi.org/10.3390/bs12120472

Methodology. Methodology is divided into several chapters and subchapters, and in my opinion it is not wise. E.g., subchapters 2.2.1 and 2.2.2 contain one sentence. In chapter 2.4, the authors indicate that “A thorough full-text reading of all 59 selected articles was performed <…>”; however, it is not clear how did they get 59 articles (there is no such number in tables and figures explaining the procedure).

Review structure. I would discuss if this chapter is necessary at all. It expands the article, but does not provide any useful information.

Discussions. I would call this part “Results”, or “Results and discussion”.

Line 403 – reference not found.

A theoretical model depicting neuromarketing in practice. This model needs to be explained more precisely. My suggestion would be to ground each arrow with sources of analyzed literature (emphasizing the number of references found). Otherwise, the model is not substantiated and can evoke many doubts, e.g., why do the authors choose using AIDA? Is it the only indicator of consumer behavior? To my opinion this chapter is not necessary.

Round 2

Reviewer 4 Report

Comments and Suggestions for Authors

Dear Authors, thank you for addressing my comments.